# The Roots of *Neorautanenia mitis* (A. Rich) Verdcourt: Further Evidence of Its Antidiarrhoeal Activity

**DOI:** 10.3390/molecules28020673

**Published:** 2023-01-09

**Authors:** Christiana J. Dawurung, Joy G. Usman, Jurbe G. Gotep, Stephen G. Pyne

**Affiliations:** 1School of Chemistry and Molecular Bioscience, Faculty of Science Medicine and Health, University of Wollongong, Wollongong, NSW 2522, Australia; 2Department of Veterinary Physiology, Biochemistry and Pharmacology University of Jos, Jos 930001, Nigeria; 3Drug Development Division, National Veterinary Research Institute (NVRI) Vom, Jos 106104, Nigeria

**Keywords:** medicinal plants, antidiarrhoeal activity, in vivo, traditional medicine, Nigerian medicinal plants

## Abstract

Despite the current management options and therapeutics used in the treatment of diarrhoea, in Africa and Asia, diarrhoea remains a major concern, especially in children under the age of 5 years. Traditional knowledge of medicinal plants used in the management of diarrhoea symptoms can be explored for their efficacy. In Nigeria, the TMPs (Traditional Medicine Practitioners) have, over the years, employed medicinal plants in the management of diarrhoea symptoms. In our current and previous studies, we aimed at validating the effectiveness of *Neorautanenia mitis* in the management of diarrhoea as claimed by the TMPs. Out of the 20 compounds isolated from *N. mitis*, the compounds neodulin, pachyrrhizine, neotenone and dolineone were the most abundant, and in this study, neodulin showed a pronounced relaxation of the rhythmic contraction of the isolated rabbit jejunum in an organ bath in a concentration-dependent manner, with a complete relaxation at 60 µg/mL. Neotenone and dolineone showed a dose-dependent inhibition of defecation of 65.07%, and 50.01%, respectively, at 20 mg/kg in a castor-oil-induced diarrhoea model. This is a strong indication that compounds from *N. mitis* possess antidiarrhoeal properties, thereby giving credence to its traditional usage in diarrhoea therapy, and therefore validating its antidiarrhoeal activity and its being worthy of further investigation.

## 1. Introduction

Diarrhoea is a global health problem and a common symptom in many diseases in both humans and animals. It is defined as an increased volume, fluidity, or frequency of faecal discharge compared with the normal stool and is characterised by abdominal pain. When diarrhoea persists, it can result in dehydration due to the loss of electrolytes, which can be fatal without intervention [1,2]. Diarrhoea can be caused by a range of microorganisms, including, viral, bacterial, parasitic, fungal, and protozoan infections. Other causes include the side effects of drugs, malnutrition, and physiological disorders. In developing countries, the periodic outbreaks of severe secretory diarrhoea caused by *Vibrio cholerae* and rota virus result in high morbidity and mortality, especially in children under the age of 5 years. Improvements in diarrhoea case management saw the mortality rate decrease by 20.8% over the period from 2005 to 2015 [3]. It was rated as the eighth-leading cause of death across all ages and the fifth-leading cause among children under the age of 5 years, causing more than 1.6 million deaths globally. Currently, it is the second-leading cause of death among children under 5 years [4,5]. In Nigeria alone, 331.3 per 100,000 children died in 2016 due to diarrhoea [6]. In 2019, 18% of children under the age of 5 years died from the disease [4,5]. Although sub-Saharan Africa and East Asia account for the highest incidence in severe disease burden, in developed countries, however, diarrhoea remains one of the common reasons for hospital consultations, which leads to discomfort and a loss of productivity [2,6,7,8,9,10,11,12,13,14]. Because diarrhoea is a common symptom in many disease conditions, it is encountered regularly; therefore, it requires immediate attention due to the consequent loss in body fluid. Medicinal plants have been employed by traditional medicine practitioners (TMPs), crop/livestock farmers, and people living in rural communities in many African countries to treat the symptoms of diarrhoea in humans and domesticated animals [15]. Information-gathering on effective medicinal plants in these communities can result in the discovery of new lead compounds for the future development of newer drugs for effective and specific diarrhoea management [16,17,18]. *Neorautanenia mitis* is used by TMPs in Nigeria for the management of diarrhoea, and in our previous studies of the crude extract of its roots, we demonstrated its antidiarrhoeal potential using an in vivo model [17,18]. This present study further validates the antidiarrhoeal activity of *N. mitis* through the evaluation of its isolated compounds on antimotility and antisecretory models in ex vivo and in vivo studies, respectively.

## 2. Results 

### 2.1. Isolation of Compounds from the Crude Extract of N. mitis

Compounds were isolated from the aqueous and DCM extracts of the root of *N. mitis.* The DCM root extract yielded 20 pure compounds which were: neoduleen (**1**), neodulin (**2**), ferulic acid (**3**), ambonane (**4**), stigmasterol (**5**), pachyrrhizine (**6**), neotenone (**7**), 7-methoxy-3-(6-methoxybenzo[d][1,3]dioxol-5-yl) chroman-4-one (**8**), 12a-hydroxydolineone (**9**), dolineone (**10**), (−)-2-isopentenyl-3-hydroxy-8-9-methylenedioxypterocarpan (**11**), nepseudin (**12**), neorautenol (**13**), isoneorautenol (**14**), (−)-2-hydroxypterocarpin (**15**), rotenone (**16**), 12a-hydroxyrotenone (**17**), dehydroneotenone (**18**), rautandiol A (**19**), and rautandiol B (**20**), as described previously [16]. The aqueous extract from the root extract of *N. mitis* yielded four compounds: **2**, **6**, **7**, and **18**. (Figure 1)

### 2.2. Studies on Isolated Rabbit Jejunum

The contractile effect was observed upon administration of the muscarinic receptor agonist acetylcholine (Ach). Following the administration of compound **2**, there was a pronounced relaxation of the contraction of the jejunum in a concentration-dependent manner, with a complete relaxation at 60 µg/mL (Figure 2)

Compounds **6**, **7**, and **10** (Figure 3, Figure 4 and Figure 5) showed moderate activity in abolishing rhythmic contractions in a concentration-dependent manner.

### 2.3. Inhibition of Castor-Oil (CO)-Induced Diarrhoea in Albino Rats

In this study, compounds isolated from the crude extract of *N. mitis,* including compounds **2**, **6**, **7,** and **10** were tested on castor-oil-induced diarrhoea in albino rats to identify the compound(s) that are responsible for the inhibition of CO-induced diarrhoea shown by the crude extract from our previous study. Compounds **7** and **10** showed a dose-dependent inhibition of defecation (see Table 1).

The percentage inhibition exhibited by these compounds at 20 mg/kg dose was 65.07 and 50.01%, respectively. Compounds **2** and **6** did not give a dose-dependent response, but it is interesting to note that at 5 mg/kg, compound **6** gave a significant (*p* ≤ 0.05) inhibition of defecation, at 74.96%, comparable to the standard drug loperamide. (Table 1)

### 2.4. Motility Test

Since compound **2** inhibited rhythmic contractions of the isolated jejunum, it was further tested for its ability to inhibit intestinal movement in rats. The result showed no significant change in the transit of activated charcoal in the GIT of rats treated with compound **2** at all dose levels when compared to that of rats treated with distilled water (Table 2). 

## 3. Discussion

Compounds **2**, **6**, **7**, and **10** were the most abundant and were also found to be active CFTR (cystic fibrosis transmembrane conductance regulator) inhibitors [18] Therefore, in this study, these compounds were further evaluated for their ability to cause the relaxation of isolated rabbit jejunum and the inhibition of castor-oil-induced diarrhoea in albino rats. These two models were designed to give further insight into the antidiarrhoeal activity of compounds from *N. mitis*.

The isolated rabbit jejunum exhibits spontaneous rhythmic contraction, which it makes possible for substances that could relax or contract smooth muscles to be identified. The contraction/relaxation effects are brought about by different mechanisms, such as stimulation of receptors, opening of ion channels, or synthesis of nitric oxide [19]. 

The complete relaxation of the jejunum by compound **2** is clearly an indication of its ability to cause the strong inhibition of gastric motility, an effect that is required in the management of diarrhoea [16,19]. Although compounds **6**, **7,** and **10** caused moderate relaxation of the jejunum, their combined activity may be responsible for the antidiarrhoeal activity recorded by the crude aqueous extract [17]. In a similar study carried out by Vongtau et al. 2000 [20], the aqueous root extract of *N. mitis* was shown to inhibit normal rhythmic activity in oxytocin-induced contraction in isolated rat uterus and acetylcholine-induced contraction in isolated rat jejunum. The study suggested that the active principle responsible for this activity may be the reason *N. mitis* is used traditionally in the treatment of dysmenorrhoea and inflammatory conditions [16].

In our earlier study, the crude extract of *N. mitis* showed inhibitory activity against CO-induced diarrhoea in albino rats [17]. The active compound in castor oil is ricinoleic acid; it has several mechanisms of action on the gastrointestinal tract. It inhibits the activity of the sodium-potassium pump, and interferes with oxidative mechanisms in the intestine to cause irritation of the intestinal walls, leading to diarrhoea. Its direct inflammatory or irritating effect on the mucosal lining of the intestine leads to the release of prostaglandins. This, in turn, affects the intestinal electrolyte balance or transport and causes an increase in membrane permeability of the mucosal cells of the intestine, leading to diarrhoea [21].

The ability of compounds **7** and **10** to cause inhibition of defecation indicates they may possess antidiarrhoeal activity by slowing intestinal motility or by reducing the irritation or inflammation caused by castor oil; this effect enhances the reabsorption of fluid from the GIT and thereby reduces fluid and electrolyte loss caused by diarrhoea. This effect is likened to the mode of action of the control drug loperamide. This drug inhibits Ach and prostaglandins release, thereby slowing intestinal motility and altering fluid and electrolyte movement across the GIT by inhibiting peristaltic activity through a direct effect on the circular and longitudinal muscles of the intestinal wall [22,23].

The lower percentage of inhibition of defecation shown by compounds **2** and **6** when compared to loperamide (85.01%) may be due to formulation and stability factors. Since the test compounds are not in drug formulations, they may therefore be prone to degradation in the stomach, thereby reaching the intestine in an insufficient amount to trigger the desired action.

In the motility studies, compound **2** showed no activity; this observation was not expected because it seems contrary to the complete inhibition of rhythmic contraction of isolated jejunum observed when compound **2** was added to the isolated rabbit jejunum in the ex vivo experiment using the organ bath (Figure 2). This disparity may be due to the inability of the compound to reach the intestine in sufficient concentration because of possible solubility problems or because of degradation in the stomach, which may be due to formulation and stability factors since the test item was not in drug formulation.

## 4. Materials and Methods

### 4.1. Extraction and Isolation of Compounds from N. mitis

As reported in our previous study [18], the roots of *N. mitis* were collected from Kabwir in Kanke Local Government Area of Plateau State, Nigeria. They were washed with water, shredded into small pieces, and dried in an oven at 45 °C for 2 days. The dried root was pulverized using a mortar and pestle. About 1000 g of the powdered material was successively extracted for 72 h with dichloromethane (DCM, 4 L) and then ethanol (4 L). The extracts were filtered through a sieve (150 µm), a cotton plug, and then filter paper (Whatmann no. 1). The filtrates were dried under a constant stream of air provided by a laboratory electric fan overnight to obtain dark-brown solids for both extracts. The yields for the DCM and ethanol extracts were 15 g (1.5%) and 14.1 g (1.4%), respectively. Both extracts showed similar TLC profiles. A portion of the DCM extract (14 g) was separated by column chromatography (CC) over silica gel using a gradient system from ethyl acetate (EtOAc)/hexanes (1:9) to 100% EtOAc to yield 40 fractions; these were combined based on their similarities by TLC and NMR analysis to afford 12 fractions. These fractions were further separated using CC (normal phase and Sepahadex LH20- Sigma Aldrich, St. Louis, MO, USA), PTLC, and crystallization to obtain 20 pure compounds (see Appendix A).

The roots of *N. mitis* were also extracted with distilled water [18]. About 1 g of the dried aqueous extract was dissolved in chloroform, filtered, and evaporated using a rotary evaporator to obtain a yield of 291.1 mg of the dried, dark-brown solid extract. This whole amount was separated by CC over silica gel using a gradient system from EtOAc/hexanes (2:8) to 100% EtOAc to give eight fractions. These fractions were further separated using CC to obtain four pure compounds [18].

### 4.2. Isolated Rabbit Jejunum Test

A rabbit jejunum was used to determine the effects of the purified compounds on the tone of gastrointestinal smooth muscles. The rabbit, weighing 3.2 g, was sacrificed by cervical dislocation and exsanguination. The abdomen was opened, and the jejunum was identified by following it back to the stomach. The small intestine was cut at a point not less than 10 cm from the stomach and not up to the caecum. The jejunum was cut into 2–3 cm pieces and placed in a Petri dish containing Tyrode’s solution. The tissue was tied at the two ends using a thread. It was tied to a tissue hook mounted in a 25 mL Panlab automatic organ bath at the lower end and a force transducer at the other end. A load of 1 g tension was applied to the tissue, and it was aerated using 95% oxygen and 5% carbon dioxide. The tissue was allowed to stabilize for 30 min; thereafter, 0.5 mL acetylcholine 10^−3^ g/mL was added to the organ bath, and the response was recorded. The tissue was then washed, and the test compounds were added cumulatively to obtain 30, 60, and 90 µg/mL final bath concentrations, and the response was recorded for each concentration [24,25,26].

### 4.3. Castor-Oil-Induced Diarrhoea 

For each pure compound tested, 9 rats of both sexes, weighing between 90–110 g, were used for the experiments. The rats were fasted for 12 h before the commencement of the experiment, having access to water only. The rats were randomly allocated into 3 groups, with each containing 3 rats. Groups 1, 2, and 3 were given graded doses (5, 10, and 20 mg/kg) orally of the pure compounds reconstituted and well-dissolved in distilled water. A total of 6 rats divided into 2 groups of 3 rats each were used as a control; rats in the first group were given distilled water (10 mL/kg), while those in the second group were given loperamide (10 mg/kg) orally. The rats were then housed singly in a perforated cage lined with white blotting paper. A total of 1 h after the above treatment, all the rats in the groups were given 1 mL of castor oil orally. The rats were observed for 5 h for watery (wet) or unformed faeces. The watery faeces from each rat were counted hourly for 5 h. At the end of the experiment, the group mean faeces was obtained, and the percentage of protection was calculated using the formula [17,27].
$$Protection=\frac{Mean\;unformed\;faeces\;of\;water\;control-treatment}{Mean\;unformed\;faeces\;of\;water\;control}\times 100$$

### 4.4. Gastrointestinal Transit of Activated Charcoal (Motility Test)

Fifteen rats of both sexes, weighing between 90 and 110 g, were used for the experiment. The rats were fasted for 16 h before the commencement of the experiment but were allowed access to water. They were randomly divided into 5 groups of 3 rats each. Groups 1, 2, and 3 were treated orally with graded doses (5 mg/kg, 10 mg/kg, and 20 mg/kg) of the test compounds reconstituted in distilled water. Groups 4 and 5 were the control groups and were treated with 5 mg/kg and 10 mL/kg of atropine sulphate and distilled water, respectively. A total of 30 min after the treatments, 1 mL of 5% activated charcoal suspension in a 10% aqueous solution of acacia gum powder was given orally to each rat. After 30 min of administering the activated charcoal, the rats were humanely sacrificed, and the abdomens were opened to access the intestines. The distance travelled by the charcoal meal from the pylorus was measured and expressed as a percentage of the total length of intestine from the pylorus to the caecum [17,21].
$$\begin{array}{c} \%Intestinal\;Transit=\frac{Distance\;Travelled\;by\;Charcoal\;Meal}{Total\;Length\;of\;Intestine\;from\;Pylorus\;to\;Caecum}\times 100 \end{array}$$

### 4.5. Statistical Analysis

The data were presented as mean ± standard deviation. The difference between the means of different treatments was determined by an analysis of variance using SPSS version 20, IBM^®^ SPSS^®^ (New York, NY, USA). *p* < 0.05 was considered to be statistically significant.

## 5. Conclusions

Compound **2** showed a significant ability to relax the rhythmic contraction of the isolated jejunum, indicating its ability to reduce GIT motility and ultimately diarrhoea. It also gave a higher inhibitory activity against castor-oil-induced diarrhoea in rats. Further investigation on the formulation stability of this compound may improve its inhibitory activity against castor-oil-induced diarrhoea and antimotility activity in in vivo studies. The results from four independent models in our collective studies on *N. mitis*, including cAMP-dependent Cl^−^ channel, TMEM16A (Ca^2+^-activated Cl^−^ channel) [18], CO-induced diarrhoea, and the isolated tissue technique (organ bath) brought to light the antisecretory and antimotility potential of *N. mitis,* thereby giving credence to its traditional usage in diarrhoea therapy, and therefore validating its antidiarrhoeal activity. 

## Figures and Tables

**Figure 1 molecules-28-00673-f001:**
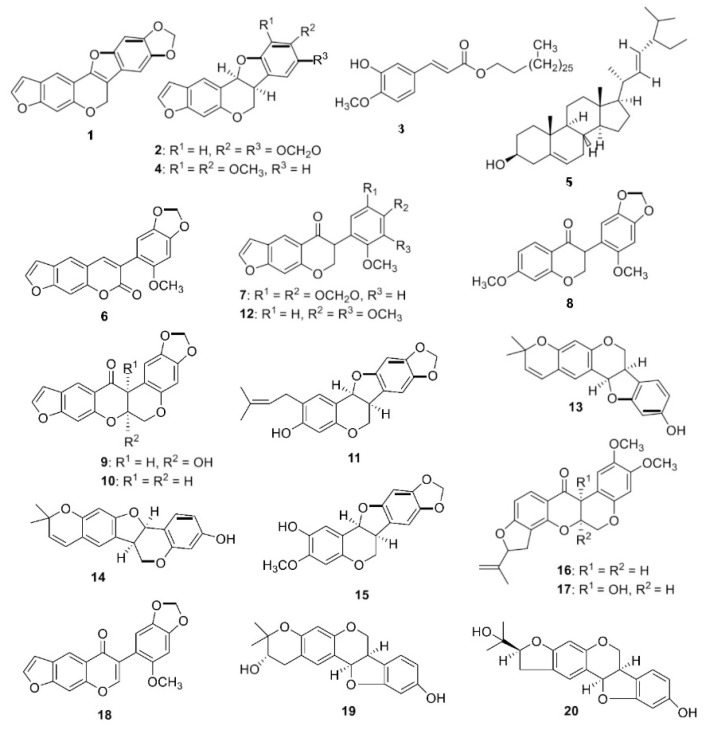
Compounds isolated from the DCM and aqueous root extracts of *N. mitis* [18].

**Figure 2 molecules-28-00673-f002:**
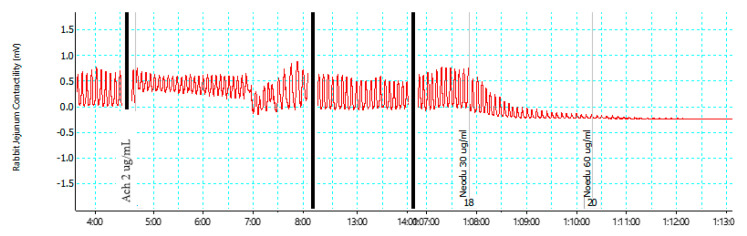
Effect of compound **2** on isolated rabbit jejunum.

**Figure 3 molecules-28-00673-f003:**
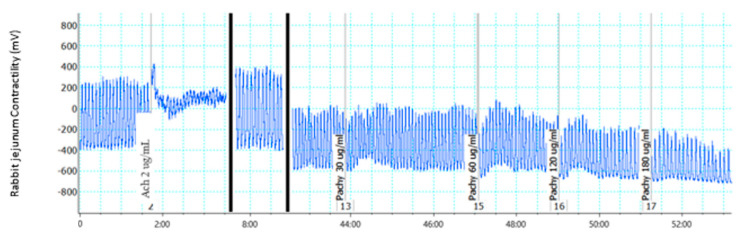
Effect of compound **6** on isolated rabbit jejunum.

**Figure 4 molecules-28-00673-f004:**
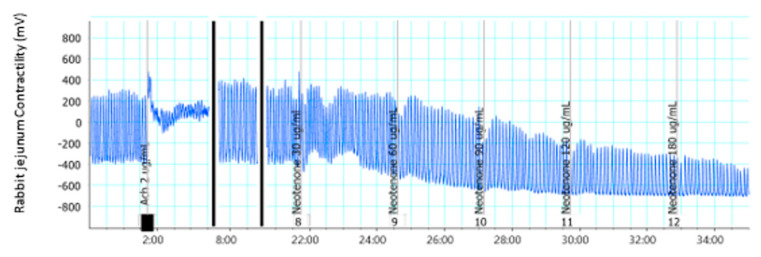
Effect of compound **7** on isolated rabbit jejunum.

**Figure 5 molecules-28-00673-f005:**
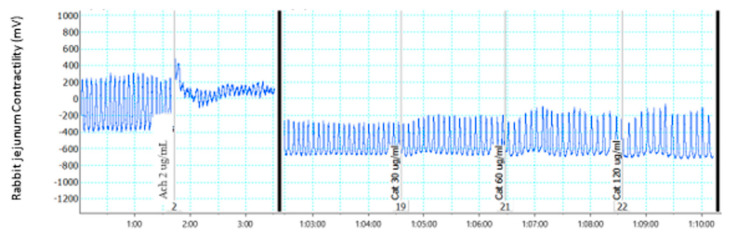
Effect of compound **10** on isolated rabbit jejunum.

**Table 1 molecules-28-00673-t001:** Percentage inhibition of defecation of rats treated with compounds **2**, **6**, **7,** and **10**.

Treatments	Mean Defecation over 6 h	Percentage Inhibition of Defecation
Castor oil + Water 10 mL/kg	6.67 ± 1.67	-
Castor oil + 5 mg/kg (**2**)	6.00 ± 2.31	10.04
Castor oil + 10 mg/kg (**2**)	4.67 ± 0.33	29.99
Castor oil + 20 mg/kg (**2**)	5.67 ± 1.20	14.99
Castor oil + Loperamide 10 mg/kg	1.00 ± 0.58 ^a^	85.01

Castor oil + Water 10 mL/kg	6.67 ± 1.67	-
Castor oil + 5 mg/kg (**6**)	1.67 ± 0.88 ^a^	74.96
Castor oil + 10 mg/kg (**6**)	4.00 ± 1.53	40.03
Castor oil + 20 mg/kg (**6**)	3.00 ± 0.58	55.02
Castor oil + Loperamide 10 mg/kg	1.00 ± 0.58 ^a^	85.01

Castor oil + Water 10 mL/kg	6.67 ± 1.67	-
Castor oil + 5 mg/kg (**7**)	4.67 ± 0.88	29.99
Castor oil + 10 mg/kg (**7**)	4.00 ± 1.53	40.03
Castor oil + 20 mg/kg (**7**)	2.33 ± 0.88	65.07
Castor oil + Loperamide 10 mg/kg	1.00 ± 0.58 ^a^	85.01

Castor oil + Water 10 mL/kg	6.67 ± 1.67	-
Castor oil + 5 mg/kg (**10**)	4.00 ± 0.58	40.03
Castor oil + 20 mg/kg (**10**)	3.33 ± 0.88	50.07
Castor oil + Loperamide 10 mg/kg	1.00 ± 0.58 ^a^	85.01

Values are expressed as mean ± SEM (*n* = 3). ^a^ = *p* ≤ 0.05 significantly different when compared with the distilled water group.

**Table 2 molecules-28-00673-t002:** Effect of neodulin (Compound **2**) on gastrointestinal transit of activated charcoal meal in rats.

Treatment	% Intestinal Transit
5 mg/kg	71.88 ± 1.26
10 mg/kg	70.58 ± 4.08
20 mg/kg	73.52 ± 6.24
Atropine 5 mg/kg	44.90 ± 5.38 **
Distilled Water	68.68 ± 9.02

Values are expressed as mean ± SD, ** = *p* = 0.001.

## Data Availability

The data presented in this study are available upon request from the corresponding author.

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
