# Peer review of "The Roots of *Neorautanenia mitis* (A. Rich) Verdcourt: Further Evidence of Its Antidiarrhoeal Activity"

_molecules, 2023, doi:10.3390/molecules28020673_

Round 1

Reviewer 1 Report

The overall article was nice and excellent research work. I have a few suggestion 

1. If the authors will add some computational work like molecular docking, molecular dynamics, etc. Then it will be better to make the research more strong.

2. Authors should go through the English editing that makes the article more readable and enjoyable for the reader.

I think after this minor work that article will be perfect to publish in this renowned journal.

Author Response

The Roots of Neorautanenia mitis (A. Rich) Verdcourt; Further evidence on its anti-diarrheal activity”

Response to Reviewer 1

Comments and Suggestions for Authors

The overall article was nice and excellent research work. I have a few suggestion;

1. If the authors will add some computational work like molecular docking, molecular dynamics, etc. Then it will be better to make the research more strong.

Response: Molecular docking and molecular dynamics are quite interesting areas of interest and the authors will consider them in their next study design.

2. Authors should go through the English editing that makes the article more readable and enjoyable for the reader.

Response:

Line 16 - 'we aim at' changed to 'we aimed at'

Line 25 - 'it is' was deleted

Line 49 - 'regularly and so it requires' is replaced with 'regularly, therefore, it requires'

Line 53 - 'the' is deleted

Line 56 - 'the root' is replaced by 'its root'

Line 82 - 'more' was deleted

Line 94 - ' from our previous study' was added

Line 116 - 'are believed' was replaced by 'were designed'

Line 125 - ' may be responsible for the activity recorded by the crude aqueous extract which showed antidiarrheal activity' was replaced by 'may be responsible for the antidiarrheal activity recorded by the crude aqueous extract'

Line 141 - 'to' was added

Line 143 - 'effect' was added

Line 144 - 'reducing' was replaced by 'reduces'

Line 144 - 'may be like the' is replaced by 'This effect is likened to'

Line 156 - 'because of the inability' was replaced by 'due to the inability'

Line 220 - 'travelled' was corrected to 'traveled'

I think after this minor work that article will be perfect to publish in this renowned journal.

The authors wish to thank you for the detailed review of this work.

Reviewer 2 Report

See file attached

Author Response

The Roots of Neorautanenia mitis (A. Rich) Verdcourt; Further evidence on its anti-diarrheal activity”

Response to Reviewer 2

In this article 20 compounds were isolated from the roots of Neorautanenia mitis by extraction with dichloromethane or water in and various experiments were carried out to obtain further evidence of anti-diarrheal activity.

1. Introduction

In the Introduction the adverse effects of diarrhea and the importance of developing new leads for anti-diarrhea drugs is described adequately.

Response: N/A

2. Results

2a. The 20 compounds isolated from the roots of Neorautanenia mitis are shown in Figure 1. The representation of compounds 2,4 and 7,12 with R2=R3= OCH2O, and R1=R2= OCH2O respectively, is a bit confusing. Why not showing two compounds (4,12) more in figure 1 and showing 2 and 7 in the same way as compounds 1 and 8?

Response:

Compounds 2 and 4 have similar structure but differ in their R positions, Compounds 7 and 12 also have similar structure but differ in their R positions. Since these two combinations share similar structure, we capture them together to show their similarities as a normal practice, and also to make a tidy presentation of the compounds and to save space.

2b. The description of the effect of compound 2, neodulin, complete relaxation of isolated rabbit jejunum at a dose of 30mg/mL is not clear to me from Figure 2 and the description of the test given in 4.2. Isolated Rabbit Jejunum Test. Please explain into some more detail.

Response:

Line 21 - 30µg/mL was changed to 60 µg/mL

Line 79 - Correction was made in the manuscript text as; The complete relaxation of rhythmic contraction of the jejunum was observed at 60ug/mL and not 30mg/mL

2c. Table 1 is clear as it shows both mean defecation in 6 hours and percentage inhibition of defecation. The description of the observed effects for compounds 2,6,7 and 10 is adequate but why no experiment was performed with compound 6, pachyrrhizine, at a dose scheme up to 5 mg/kg? As the authors indicate the effect of 6 at 5 mg/kg is very comparable to the effect of the well-known reference drug loperamide.

Response:

Compound 6 is being considered in for further studies in our future investigation of the roots of N. mitis because of its significant effect at low dose.

3. Discussion

P6 line 6 typo: “their combine activity” should be “their combined activity”. The Discussion of the possible effects of the 4 compounds is adequate but still mainly speculative as further research is needed.

Response:

Line 125- Corrected to 'their combined activity'

4. Materials and Methods

The description of the 4.1. Extraction and isolation of compounds from N. mitis, 4.2. Isolated Rabbit Jejunum Test, 4.3. Castor oil induced diarrhea, 4.4. Gastrointestinal Transit of Activated Charcoal (Motility Test) and 4.5. Statistical analysis is adequate.

Response:

N/A

5. Conclusion

The Conclusion in itself is well presented but contains elements that might be better described in 3. Missing is the possible combined activity of compounds 2,6,7 and 10. Though usually studies are performed on single potential drugs, it seems worthwhile to study the combined activity as well as it might improve the effectivity of diarrhea treatment considerably.

Response:

In our future study design, these compounds will be considered for formulation to improve their possible stability in-vivo. Your suggestion on testing their combine effect is valid and will be considered in the future.

The Authors wish to appreciate you for the detailed review to this work.